# The Medical Segmentation Decathlon

Michela Antonelli [1,42✉], Annika Reinke [2,3,4,42], Spyridon Bakas [5,6,7], Keyvan Farahani[8], Annette Kopp-Schneider [9], Bennett A. Landman [10], Geert Litjens [11], Bjoern Menze [12], Olaf Ronneberger[13], Ronald M. Summers[14], Bram van Ginneken[11], Michel Bilello[5], Patrick Bilic[15], Patrick F. Christ[15], Richard K. G. Do [16], Marc J. Gollub[16], Stephan H. Heckers[17], Henkjan Huisman [11], William R. Jarnagin[18], Maureen K. McHugo[17], Sandy Napel [19], Jennifer S. Golia Pernicka [16], Kawal Rhode[1], Catalina Tobon-Gomez[1], Eugene Vorontsov[20], James A. Meakin[11], Sebastien Ourselin[1], Manuel Wiesenfarth[9], Pablo Arbeláez[21], Byeonguk Bae [22], Sihong Chen[23], Laura Daza[21], Jianjiang Feng [24], Baochun He[25], Fabian Isensee[26], Yuanfeng Ji[27], Fucang Jia [25], Ildoo Kim[28], Klaus Maier-Hein [29,30], Dorit Merhof [31,32], Akshay Pai[29,33], Beomhee Park[22], Mathias Perslev [33], Ramin Rezaiifar[34], Oliver Rippel[31], Ignacio Sarasua[35], Wei Shen[36], Jaemin Son[22], Christian Wachinger[35], Liansheng Wang[27], Yan Wang[37], Yingda Xia[38], Daguang Xu[39], Zhanwei Xu [24], Yefeng Zheng [23], Amber L. Simpson[40], Lena Maier-Hein[2,3,4,41,43] & M. Jorge Cardoso [1,43]

International challenges have become the de facto standard for comparative assessment of image analysis algorithms. Although segmentation is the most widely investigated medical image processing task, the various challenges have been organized to focus only on specific clinical tasks. We organized the Medical Segmentation Decathlon (MSD)—a biomedical image analysis challenge, in which algorithms compete in a multitude of both tasks and modalities to investigate the hypothesis that a method capable of performing well on multiple tasks will generalize well to a previously unseen task and potentially outperform a custom-designed solution. MSD results confirmed this hypothesis, moreover, MSD winner continued generalizing well to a wide range of other clinical problems for the next two years. Three main conclusions can be drawn from this study: (1) state-of-the-art image segmentation algorithms generalize well when retrained on unseen tasks; (2) consistent algorithmic performance across multiple tasks is a strong surrogate of algorithmic generalizability; (3) the training of accurate AI segmentation models is now commoditized to scientists that are not versed in AI model training.

A full list of author affiliations appears at the end of the paper.

Machine learning is beginning to revolutionize many fields of medicine, with success stories ranging from the accurate diagnosis and staging of diseases[1], to the early prediction of adverse events[2] and the automatic discovery of antibiotics[3]. In this context, a large amount of literature has been dedicated to the automatic analysis of medical images[4]. Semantic segmentation refers to the process of transforming raw medical images into clinically relevant, spatially structured information, such as outlining tumor boundaries, and is an essential pre-requisite for a number of clinical applications, such as radio-therapy planning[5] and treatment response monitoring[6]. It is so far the most widely investigated medical image processing task, with about 70% of all biomedical image analysis challenges dedicated to it[7]. With thousands of algorithms published in the field of biomedical image segmentation per year[8], however, it has become challenging to decide on a baseline architecture as starting point when designing an algorithm for a new given clinical problem.

International challenges have become the de facto standard for comparative assessment of image analysis algorithms given a specific task[7]. Yet, a deep learning architecture well-suitable for a certain clinical problem (e.g., segmentation of brain tumors) may not necessarily generalize well to different, unseen tasks (e.g., vessel segmentation in the liver). Such a "generalizable learner", which in this setting would represent a fully-automated method that can learn any segmentation task given some training data and without the need for human interven-tion, would provide the missing technical scalability to allow many new applications in computer-aided diagnosis, biomarker extraction, surgical intervention planning, disease prognosis, etc. To address this gap in the literature, we proposed the concept of the Medical Segmentation Decathlon (MSD), an international challenge dedicated to identifying a general-purpose algorithm for medical image segmentation. The com-petition comprised ten different data sets with various chal-lenging characteristics, as shown in Fig. 1. Two subsequent phases were presented to participants, first the development phase serving for model development and including seven open training data sets. Then, the mystery phase, aiming to investi-gate whether algorithms were able to generalize to three unseen segmentation tasks. During the mystery phase, participants were allowed to submit only one solution, able to solve all problems without changing the architecture or hyperparameters.

The contribution of this paper is threefold: (1) We are the first to organize a biomedical image analysis challenge in which algorithms compete in a multitude of both tasks and modalities. More specifically, the underlying data set has been designed to feature some of the representative difficulties typically encountered when dealing with medical images, such as small data sets, unbalanced labels, multi-site data and small objects. (2) Based on the MSD, we released the first open framework for benchmarking medical segmentation algorithms with a specific focus on generalizability. (3) By monitoring the winning algo-rithm, we show that generalization across various clinical applications is possible with one single framework.

In the following, we will show the MSD results in "Results", in which we present the submitted methods and rankings based on the Dice Similarity Coefficient (DSC)[9] and the Normalized Sur-face Dice (NSD)[10] metrics as well as the results for the live challenge. We conclude with a discussion in "Discussion". The challenge design, including the mission, challenge data sets and assessment method, can be found in the "Methods". Further details including the overall challenge organization, detailed participating method descriptions and further results are pre-sented in the Supplementary Information.

## Results

**Challenge submissions.** In total, 180 teams registered for the challenge, from which 31 submitted fully-valid and complete results for the development phase. From these, a subset of 19 teams submitted final and valid results for the mystery phase. Among the methods that fulfilled all the criteria to move to the mystery phase, all methods were based on convolutional neural networks, with the U-Net[11] being the most frequently used base architecture—employed by more than half of the teams (64%). The most commonly used loss function was the DSC loss (29%), followed by the cross entropy loss (21%). Figure 2 provides a complete list of both network architectures and loss functions used in the challenge. 61% of the teams used the adaptive moment estimation (Adam) optimizer[12], while the stochastic gradient descent (SGD)[13] was used by 33% of the teams.

**Method description of top three algorithms.** In the following, the top three methods are briefly described while the remaining participating methods are described in the Supplementary Methods 2. Supplementary Table 1 further provides an overview over all methods that were submitted for the mystery phase and who provided full algorithmic information ($n = 14$ teams), including links to public repositories (when available).

The key idea of nnU-Net's method was to use a fully-automated dynamic adaptation of the segmentation pipeline, done indepen-dently for each task in the MSD, based on an analysis of the respective training data set. Image pre-processing, network topologies and post-processing were determined fully automatically and considered more important than the actual architecture[8]. nnU-Net was based on the U-Net architecture[11] with the following modifications: the use of leaky ReLU, instance normalization and strided convolutions for downsampling[8]. It further applied a combination of augmentation strategies, namely affine transforma-tion, non-linear deformation, intensity transformation (similar to gamma correction), mirroring along all axes and random crop. The sum of the DSC and cross entropy loss was used, while utilizing the Adam optimizer. The method applied a purposely defined ensembling strategy in which four different architectures were used. The selection of the task-specific optimal combination was found automatically via cross-validation on the training set.

The key idea of NVDLMED's method was to use a fully-supervised uncertainty-aware multi-view co-training strategy[14]. They achieved robustness and generalization by initializing the model from 2D pre-trained models and using three views of networks to gain more 3D information through the multi-view co-training process. They further used a resampling strategy to cope with the differences among the ten tasks. The NVDLMED team utilized a 3D version of the ResNet[15] with anisotropic 3D kernels[14]. The team further applied a combination of augmenta-tion strategies, namely affine transformation, geometric left-right flip and random crop. The DSC loss and the SGD optimizer were employed. NVDLMED ensembled three models, each trained on a different view (coronal, saggital and axial).

The key idea of K.A.V.athlon's method was a generalization strategy in the spirit of AutoML[16]. The process was designed to train and predict automatically using given image data and description without any parameter change or intervention by a human. K.A.V.athlon's method was based on a combination of the V-Net and U-Net architectures with the addition of a Squeeze-and-Excitation (SE) block and a residual block. The team further applied different types of augmentation, namely affine transformation, noise application, geometric left-right flip, random crop, and blurring. The DSC loss with a thresholded ReLU (threshold 0.5) and the Adam optimizer were employed. No ensembling strategy was used.

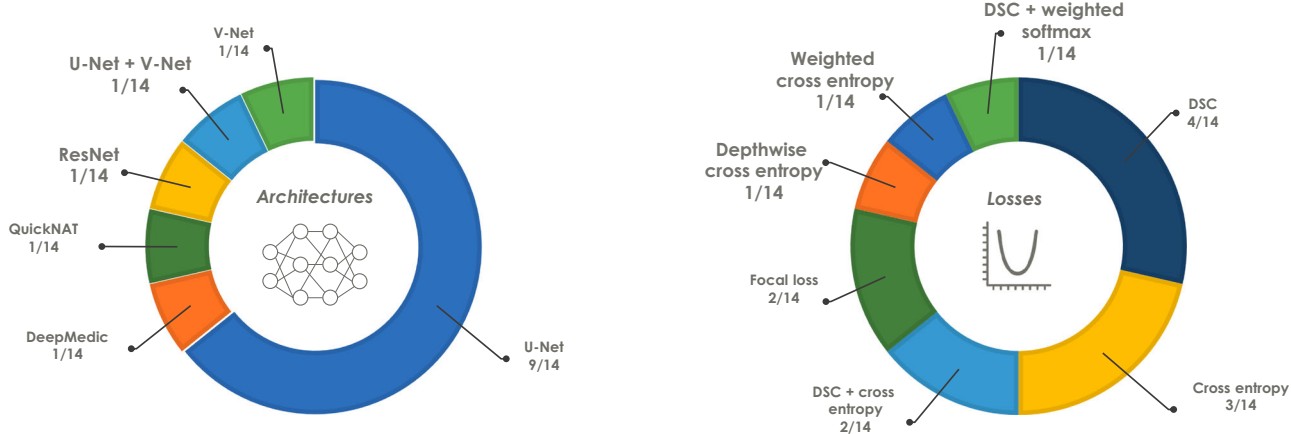

**Fig. 1 Overview of the ten different tasks of the Medical Segmentation Decathlon (MSD).** The challenge comprised different target regions, modalities and challenging characteristics and was separated into seven known tasks (blue; the development phase: brain, heart, hippocampus, liver, lung, pancreas, prostate) and three mystery tasks (gray; the mystery phase: colon, hepatic vessels, spleen). MRI magnetic resonance imaging, mp-MRI multiparametric-magnetic resonance imaging, CT computed tomography.

**Fig. 2 Base network architectures (left) and loss functions (right) used by the participants of the 2018 Decathlon challenge who provided full algorithmic information (n = 14 teams).** Network architectures: DeepMedic—Efficient multi-scale 3D CNN with fully connected CRF for accurate brain lesion segmentation[45], QuickNAT—Fully Convolutional Network for Quick and Accurate Segmentation of Neuroanatomy[44], ResNet—Deep Residual Learning for Image Recognition[15], U-Net—Convolutional Networks for Biomedical Image Segmentation[11], V-Net—Fully Convolutional Neural Networks for Volumetric Medical Image Segmentation[43]. DSC Dice Similarity Coefficient.

## Development Phase

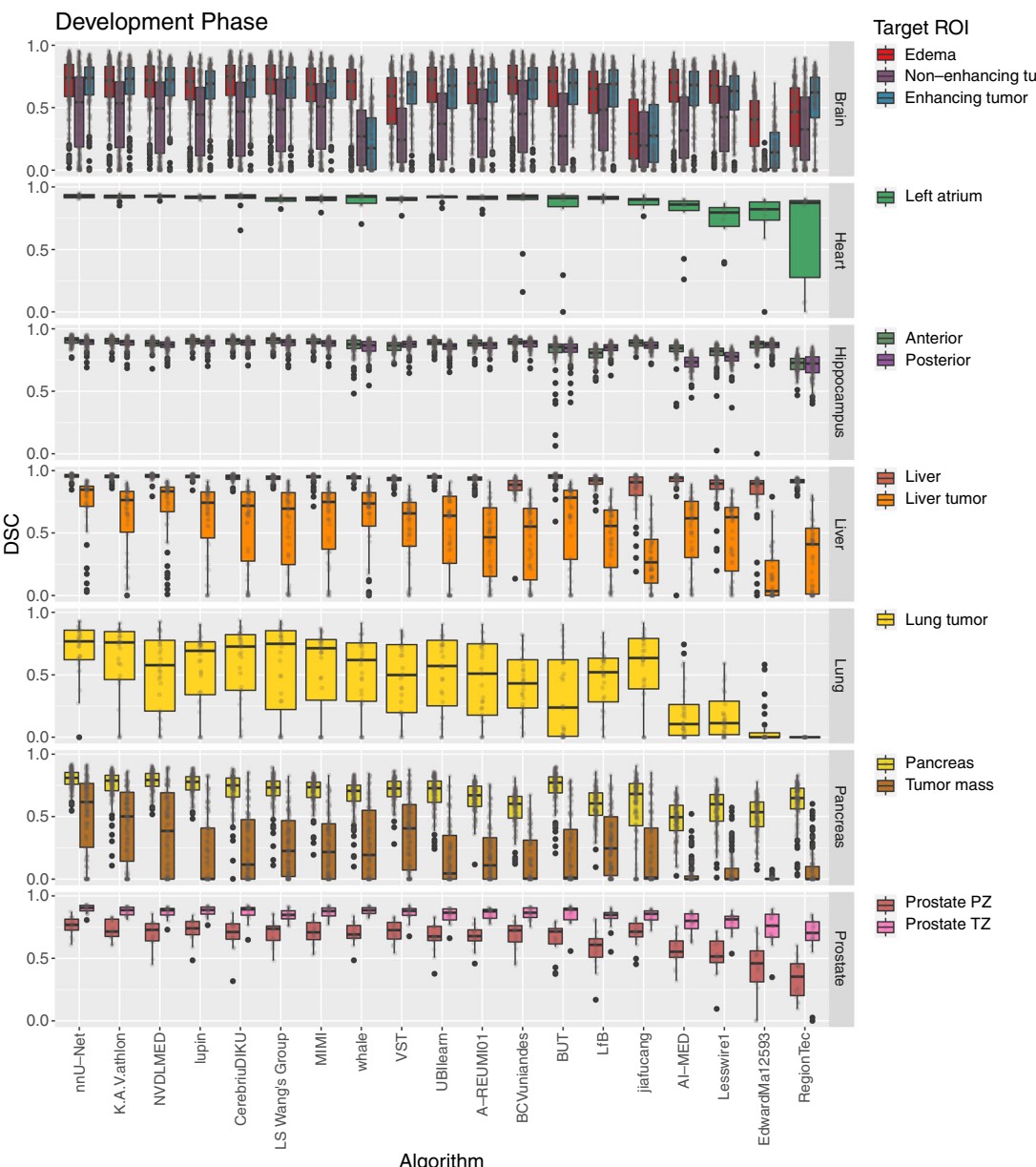

**Fig. 3 Dot- and box-plots of the Dice Similarity Coefficient (DSC) values of all 19 participating algorithms for the seven tasks (brain, heart, hippocampus, liver, lung, pancreas, prostate) of the development phase, color-coded by the target regions (edema (red), non-enhancing tumor (purple), enhancing tumor (blue), left atrium (green), anterior (olive), posterior (light purple), liver (dark orange), liver tumor (orange), lung tumor (yellow), pancreas (dark yellow), tumor mass (light brown), prostate peripheral zone (PZ) (brown), prostate transition zone (TZ) (pink)).** The box-plots represent descriptive statistics over all test cases. The median value is shown by the black horizontal line within the box, the first and third quartiles as the lower and upper border of the box, respectively, and the 1.5 interquartile range by the vertical black lines. Outliers are shown as black circles. The raw DSC values are provided as gray circles. ROI Region of Interest.

**Individual performances and rankings.** The DSC values for all participants for the development phase and the mystery phase are provided as dot- and box-plots in Figs. 3, 4, respectively. For tasks with multiple target ROIs (e.g., edema, non-enhancing tumor and enhancing tumor segmentation for the brain data set), the box-plots were color-coded according to the ROI. The distribution of the NSD metric values was comparable to the DSC values and can be found in Supplementary Figs. 1, 2.

It can be seen that the performance of the algorithms as well as their robustness depends crucially on the task and target ROI. The median of the mean DSC computed considering all test cases of a single task over all participants ranged from 0.16 (colon cancer segmentation (the mystery phase), cf. Supplementary

Table 9) to 0.94 (liver (the development phase), cf. Supplementary Table 5) and spleen segmentations (the mystery phase), cf. Supplementary Table 11). The full list of values are provided in the Supplementary Tables 2–11.

The rankings for the challenge are shown in Table 1. The winning method (nnU-Net) was extremely robust with respect to the different tasks and target regions for both phases (cf. Figs. 3, 4). Ranks 2 and 3 switched places (K.A.V.athlon and NVDLMED) for both the development and mystery phase. Figure 5 further shows the ranks of all algorithms for all thirteen target regions of the development phase (red) and all four target regions of the mystery phase in form of a box-plot. Many teams show a large variation in their ranks across target ROIs. The lowest rank difference of three

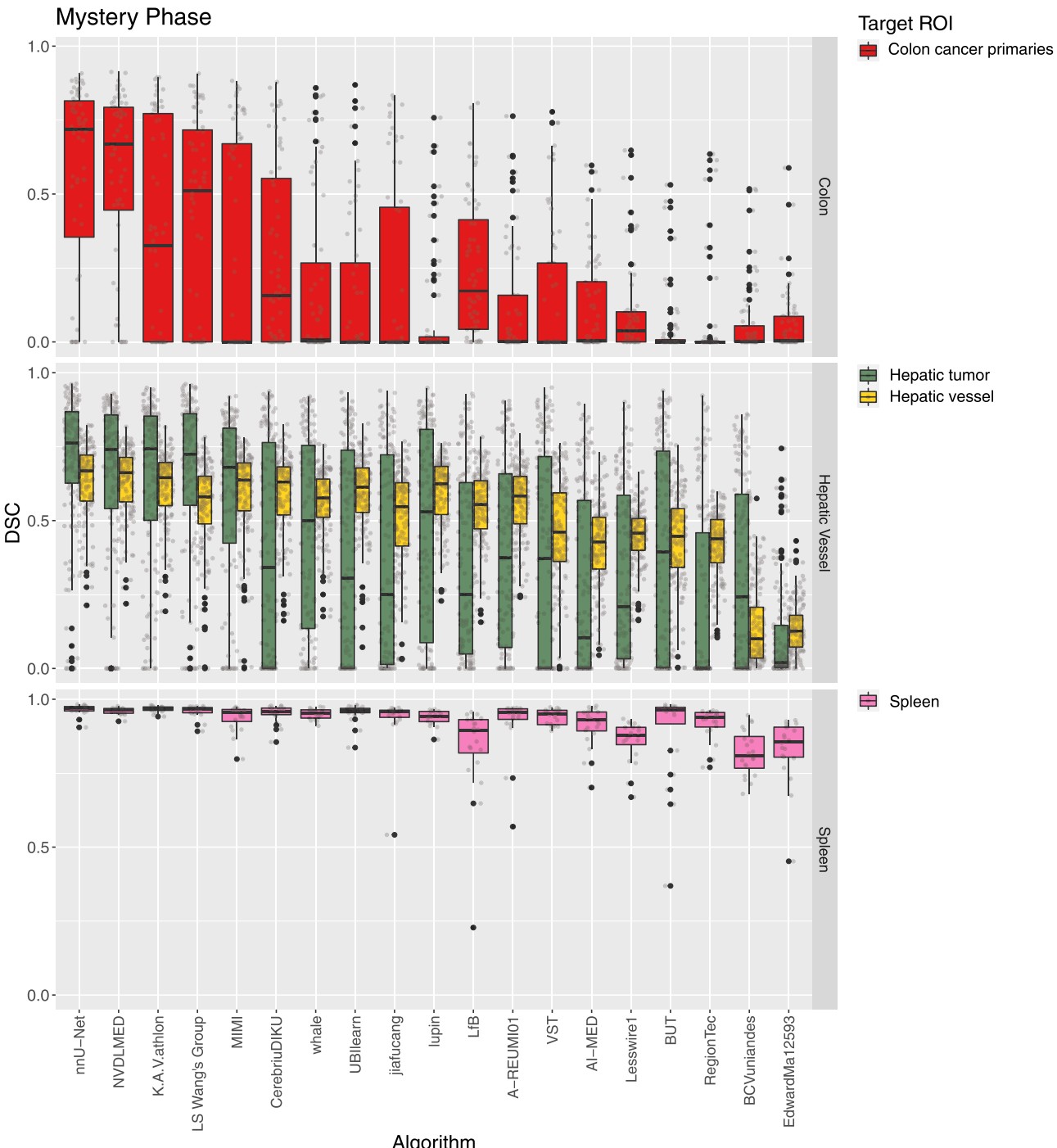

**Fig. 4 Dot- and box-plots of the Dice Similarity Coefficient (DSC) values of all 19 participating algorithms for the three tasks of the mystery phase (colon, hepatic vessel, spleen), color-coded by the target regions (colon cancer primaries (red), hepatic tumor (green), hepatic vessel (yellow), spleen (pink)).** The box-plots represent descriptive statistics over all test cases. The median value is shown by the black horizontal line within the box, the first and third quartiles as the lower and upper border of the box, respectively, and the 1.5 interquartile range by the vertical black lines. Outliers are shown as black circles. The raw DSC values are provided as gray circles. ROI Region of Interest.

ranks was achieved for team nnU-Net (minimum rank: 1, maximum rank: 4; the development phase) and the largest rank difference of sixteen ranks is obtained for team Whale (minimum rank: 2, maximum rank: 18; the development phase).

To investigate ranking robustness, line plots[17] are provided in the Supplementary Figs. 3–12 for all individual target regions, indicating how ranks change for different ranking schemes. Furthermore, a comparison of the achieved ranks of algorithms for 1000 bootstrapped samples is provided in the form of a stacked

frequency plot[17] in Supplementary Fig. 13. For each participant, the frequency of the achieved ranks is provided for every task individually. It can be easily seen from both uncertainty analyses that team nnU-Net implemented an extremely successful method that was at rank 1 for nearly every tasks and bootstrap set.

The variability of the original rankings computed for the development phase and the mystery phase and the ranking lists based on the individual bootstrap samples was determined via Kendall's $\tau$[18]. The median (interquartile range (IQR)) Kendall's $\tau$

**Table 1 Rankings for the development phase and the mystery phase, median and interquartile range (IQR) of the Dice Similarity Coefficient (DSC) values of all 19 teams.**

| The development phase | | | | The mystery phase | | | |
|---|---|---|---|---|---|---|---|
| Rank | Team ID | Median DSC | IQR DSC | Rank | Team ID | Median DSC | IQR DSC |
| 1 | nnU-Net | 0.79 | (0.61, 0.88) | 1 | nnU-Net | 0.71 | (0.58, 0.82) |
| 2 | K.A.V.athlon | 0.77 | (0.58, 0.87) | 2 | NVDLMED | 0.69 | (0.55, 0.79) |
| 3 | NVDLMED | 0.78 | (0.57, 0.87) | 3 | K.A.V.athlon | 0.67 | (0.49, 0.80) |
| 4 | Lupin | 0.75 | (0.52, 0.86) | 4 | LS Wang's Group | 0.64 | (0.46, 0.78) |
| 5 | CerebriuDIKU | 0.76 | (0.51, 0.88) | 5 | MIMI | 0.65 | (0.45, 0.75) |
| 6 | LS Wang's Group | 0.75 | (0.51, 0.88) | 6 | CerebriuDIKU | 0.56 | (0.15, 0.71) |
| 7 | MIMI | 0.73 | (0.51, 0.86) | 7 | Whale | 0.55 | (0.20, 0.68) |
| 8 | Whale | 0.65 | (0.28, 0.83) | 8 | UBIlearn | 0.55 | (0.05, 0.69) |
| 9 | VST | 0.69 | (0.39, 0.84) | 9 | Jiafucang | 0.48 | (0.04, 0.67) |
| 10 | UBIlearn | 0.72 | (0.40, 0.85) | 10 | Lupin | 0.57 | (0.19, 0.69) |
| 11 | A-REUMI01 | 0.70 | (0.42, 0.85) | 11 | LfB | 0.49 | (0.16, 0.64) |
| 12 | BCVuniandes | 0.70 | (0.42, 0.86) | 12 | A-REUMI01 | 0.51 | (0.14, 0.65) |
| 13 | BUT | 0.72 | (0.40, 0.84) | 13 | VST | 0.41 | (0.00, 0.64) |
| 14 | LfB | 0.68 | (0.43, 0.82) | 14 | AI-MED | 0.33 | (0.01, 0.52) |
| 15 | Jiafucang | 0.49 | (0.11, 0.81) | 15.5 | Lesswire1 | 0.40 | (0.08, 0.52) |
| 16 | AI-Med | 0.63 | (0.30, 0.79) | 15.5 | BUT | 0.38 | (0.01, 0.60) |
| 17 | Lesswire1 | 0.65 | (0.33, 0.79) | 17 | RegionTec | 0.29 | (0.00, 0.50) |
| 18 | EdwardMa12593 | 0.31 | (0.01, 0.69) | 18 | BCVuniandes | 0.10 | (0.01, 0.38) |
| 19 | RegionTec | 0.57 | (0.19, 0.73) | 19 | EdwardMa12593 | 0.08 | (0.01, 0.17) |

The ranking was computed as described in "Assessment of competing teams".

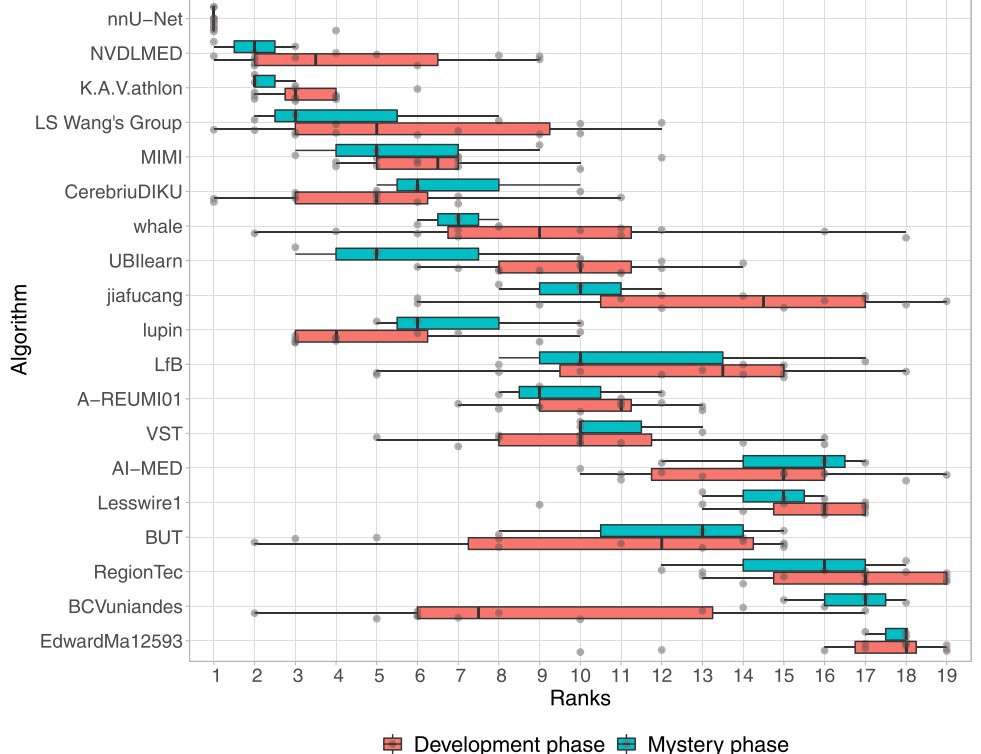

**Fig. 5 Dot- and box-plots of ranks for all 19 participating algorithms over all seven tasks and thirteen target regions of the development phase (red) and all three tasks and four target regions of the mystery phase (blue).** The median value is shown by the black vertical line within the box, the first and third quartiles as the lower and upper border of the box, respectively, and the 1.5 interquartile range by the horizontal black lines. Individual ranks are shown as gray circles.

was 0.94 (0.91, 0.95) for the colon task, 0.99 (0.98, 0.99) for the hepatic-vessel task and 0.92 (0.89, 0.94) for the spleen task. This shows that the rankings for the mystery phase were stable against small perturbations.

**Impact of the challenge winner.** In the 2 years after the challenge, the winning algorithm, nnU-Net (with sometimes minor modification) competed in a total of 53 further segmentation tasks. The method won 33 out of 53 tasks with a median rank of 1

(interquartile range (IQR) of (1;2)) in the 53 tasks[8], for example being the winning method of the famous BraTS challenge in 2020 (Team Name: MIC_DKFZ, https://www.med.upenn.edu/cbica/brats2020/rankings.html). This confirmed our hypothesis that a method capable of performing well on multiple tasks will generalize well to a previously unseen task and potentially outperform a custom-designed solution. The method further became the new state-of-the-art method and was used in several segmentation challenges by other researchers. For instance, eight nnU-Net derivatives were ranked in the top 15 algorithms of the 2019 Kidney and Kidney Tumor Segmentation Challenge (KiTS—https://kits19.grand-challenge.org/)[8], the MICCAI challenge with the most participants in the year 2019. Nine out of the top ten algorithms in the COVID-19 Lung CT Lesion Segmentation Challenge 2020 (COVID-19-20 https://covid-segmentation.grand-challenge.org/) built their solutions on top of nnU-Net (98 participants in total). As demonstrated in[19], nine out of ten challenge winners in 2020 built solutions on top of nnU-Net.

## Discussion

We organized the first biomedical image segmentation challenge, in which algorithms competed in ten different disciplines. We showed that it is indeed possible that one single algorithm can generalize over various different applications without human-based adjustments. This was further demonstrated by monitoring the winning method for 2 years to show the continuation of the generalizability to other segmentation tasks.

In the following sections, we will discuss specific aspects of the MSD challenge, namely the challenge infrastructure, data set, assessment method and outcome.

**Challenge infrastructure**. The participating teams were asked to submit their results in the form of a compressed archive to the grand-challenge.org platform. For the development phase, a fully-automated validation script was run for each submission and the leaderboard was updated accordingly. Each team was allowed to submit one solution per day. In contrast, for the mystery phase, only one valid submission per algorithm could be submitted to prevent overfitting.

Despite the above-mentioned policies, there were attempts to create multiple accounts so that a team could test their method beyond the allowed limit, a problem which was found due to result's similarity between certain accounts. Teams who were found to be evading the rules were disqualified. Identity verification and fraud detection tooling has now been added to grand-challenge.org to help organizers mitigate this problem in the future.

Possibly, a better way of controlling overfitting, or possible forms of cheating (e.g., manual refinement of submitted results[20]) would have been to containerize the algorithms using Docker containers and for inference to be run by the organizers. This approach was unfortunately not possible at the time of the organization of MSD due to the lack of computational resources to run inference on all data for all participants. Thanks to a partnership with Amazon Web Services (AWS), the grand-challenge.org platform now offers the possibility to upload Docker container images that can participate in challenges and made available to researchers for processing new scans. With the recent announcement of a partnership between NVIDIA and the MICCAI 2020 and 2021 conferences, and the increased standardization of containers, such a solution should be adopted for further iterations of the MSD challenge.

**Challenge data set**. In the MSD, we presented a unique data set, including ten heterogeneous tasks from various body parts and regions of interest, numerous modalities and challenging characteristics. MSD is the largest and most comprehensive medical image segmentation data set available to date. The MSD data set has been downloaded more than 2000 times in its first year alone, via the main challenge website (http://medicaldecathlon.com/). The data set has recently been accepted into the AWS Open-Data registry, (https://registry.opendata.aws/msd/) allowing for unlimited download and availability. The data set is also publicly available under a Creative Commons license CC-BY-SA4.0, allowing broad (including commercial) use. Due to data set heterogeneity, and usage in generalizability and domain adaptation research, it is likely to be very valuable for the biomedical image analysis community in the long term.

Regarding limitations, the MSD data set was gathered from retrospectively acquired and labeled data from many different sources, resulting in heterogeneous imaging protocols, differences in annotation procedures, and limiting the annotations to a single human rater. While the introduction of additional annotators would have benefited the challenge by allowing inter-rater reliability estimates and possibly improve the reliability of annotations, this was not possible due to restricted resources and the scale of the data. As shown in[21], several annotators are often necessary to overcome issues related to inter-observer variability. Furthermore, the data set only consists of radiological data, we can therefore only draw conclusions for this application. Other areas like dermatology, pathology or ophthalmology were not covered. Finally, one specific region from one task (the vessel annotations of liver data set) was found to be non-optimal from a segmentation point of view after the data release, we opted to follow the best practice recommendations on challenges[7, 20, 22] and not change the challenge design after it was released to participants. Note, however, that the message of this challenge would not change if the vessel data set was omitted from the competition.

**Challenge assessment**. Two common segmentation metrics have been used to evaluate the participant's methods, namely the DSC, an overlap measure, and the NSD, a distance-based metric. The choice of the right metrics was heavily discussed, as it is extremely important for the challenge outcome and interpretation. Some metrics are more suitable for specific clinical use-cases than others[23]. For instance, the DSC metric is a good proxy for comparing large structures but should not be used intensively for very small objects, as single-pixel differences may already lead to substantial changes in the metric scores. However, to ensure that the results are comparable across all ten tasks, a decision was taken to focus on the two above-mentioned metrics, rather than using clinically-driven task-specific metrics.

Comparability was another issue for the ranking as the number of samples varied heavily across all tasks and target ROIs, which made a statistical comparison difficult. We therefore decided to use a ranking approach similar to the evaluation of the popular BraTS challenge, (http://braintumorsegmentation.org/) which was based on a Wilcoxon-signed-rank pairwise statistical test between algorithms. The rank of each algorithm was determined (independently per task and ROI) by counting the number of competing algorithms with a significantly worse performance. This strategy avoided the need of similar sample sizes for all tasks and reduced the need for task-specific weighting and score normalization.

Identifying an appropriate ranking scheme is a non-trivial challenge. It is important to note that each task of the MSD data set comprised one to three different target ROIs, introducing a hierarchical structure within the data set. Starting from a significance ranking for each target ROI, we considered two

different aggregation schemes: (1) averaging the significance ranks across all target ROIs; (2) averaging the significance ranks per task (data set) and averaging those per-task ranks for the final rank. The drawback of (1) is that a possible bias between tasks might be introduced, as tasks with multiple target ROIs (e.g., the brain task with three target ROIs) would be over-weighted. We therefore chose ranking scheme (2) to avoid this issue. This decision was made prior to the start of the challenge, as per the challenge statistical analysis protocol. A post-challenge analysis was performed to test this decision, and results found that overall ranking structure remained unchanged. The first three ranks were preserved, only minor changes (1 to 2 ranks) were seen in a couple of examples at the middle and end of the rank list. As shown in Supplementary Figs. 3–12, changing the ranking scheme will typically lead to different rankings in the end, but we observed the first three ranks to be robust across various ranking variations. More complex ranking schemes were discussed among organizers, such as modeling the variations across tasks and target ROIs with a linear mixed model[24]. As explainability and a clear articulation of the ranking procedure was found to be important, it was ultimately decided to use significance ranking.

**Challenge outcome**. A total of 180 teams registered for the MSD challenge, of which only 31 teams submitted valid results for the development and 19 teams for the mystery phase. Having a high number of registrations but only a fraction of final participants is a typical phenomenon happening for biomedical image analysis challenges (e.g., the Skin lesion analysis toward melanoma detection 2017 challenge with 46/593 submissions[25], the Robust Medical Instrument Segmentation (RobustMIS) challenge 2019 with 12/75 submissions[26] or the Multi-Center, Multi-Vendor, and Multi-Disease Cardiac Segmentation (M&Ms) challenge 2020 with 16/80 submissions[27]). Many challenge participants usually register to get data access. However, teams are often not able to submit their methods within the deadline due to other commitments. Furthermore, participants may be dissatisfied with their training and validation performance and step back from the final submission. The performance of the submitted algorithms varied dramatically across the different tasks, as shown in Figs. 3, 4 and Supplementary Tables 2–11. For the development phase, the median algorithmic performance, defined as the median of the mean DSC, changed widely across tasks, with lowest being the tumor mass segmentation of the pancreas data set (0.21, Supplementary Table 7) and the highest median for the liver segmentation (0.94, Supplementary Table 5). The performance drop was much more modest for the best performing method nnU-Net (0.52 and 0.93 median DSC for the pancreas mass and liver ROI, respectively), demonstrating that methods have varying degrees of learning resiliency to the challenges posed by each task. The largest difference within one task was also obtained for the pancreas data set, with a median of the mean DSC of 0.69 for the pancreas ROI, and 0.21 for the pancreas tumor mass, which is likely explained by the very small relative intensity difference between the pancreas and its tumor mass.

In the mystery phase, colon cancer segmentation received the lowest median DSC (0.16, Supplementary Table 9), and the spleen segmentation the highest median DSC (0.94, Supplementary Table 11). Similarly to the development phase, a much smaller drop in performance (0.56 and 0.96 for colon and spleen respectively) was observed in the top ranking method. Most of the observed task-specific performances reflect the natural difficulty and expected inter-rater variability of the tasks: Liver and spleen are large organs that are easy to detect and outline[28], whereas pancreas and colon cancers are much harder to segment as

annotation experts themselves often do not agree on the correct outlines[29, 30]. We also observed that the challenging characteristics of each task (presented in Fig. 1) had some non-trivial effect on algorithmic performance, a problem which was exacerbated in lower-ranking methods. For example, some methods struggled to segment regions such as the lung cancer mass, pancreas mass, and colon cancer primaries, achieving a mean DSC below 0.1. These regions, characterized by small, non-obvious and heterogeneous masses, appear to represent a particularly challenging axis of algorithmic learning complexity. The number of subjects in the training data set (only 30 subjects for the heart task), the size and resolution of the images (large liver images and small hippocampus images), and complex region shapes (e.g., brain tumors) were not found to introduce significant inter-team performance differences. As summarized in Supplementary Fig. 13, nnU-Net was ranked first on both the development and mystery phases. Under the proposed definition of a "generalizable learner", the winning method was found to be the most generalizable approach across all MSD tasks given the comparison methodology, with a significant performance margin. The K.A.V.athlon and NVDLMED teams were ranked second and third during the development phase, respectively; their ranks were swapped (third and second, respectively) during the mystery phase. We observed small changes in team rankings between the development and mystery phases for top ranking teams; within the top 8 teams, no team changed their ranking by more than 2 positions from the development to the mystery phase. This correlation between development and mystery rankings suggest limited amount of methodological overfitting to the development phase, and that the proposed ranking approach is a good surrogate of expected task performance. We observed some algorithmic commonalities between top methods, such the use of ensembles, intensity and spatial normalization augmentation, the use of Dice loss, the use of Adam as an optimizer, and some degree of post-processing (e.g., region removal). While none of these findings are surprising, they provide evidence towards a reasonable choice of initial parameters for new methodological developments. We also observed that the most commonly applied architecture across participants was the U-Net, used by 64% of teams. Some evidence was found that architectural adjustments to the baseline U-Net approach are less important than other relevant algorithmic design decisions, such as data augmentation and data set split/ cross-validation methodology, as demonstrated by the winning methodology. Note that similar findings, albeit in a different context and applied to ResNet, have been recently observed[31].

**The years after the challenge**. Following the challenge event at MICCAI 2018, the competition was opened again for rolling submissions. This time participants were asked to submit results for all ten data sets (https://decathlon-10.grand-challenge.org/) in a single phase. In total, 742 users signed up. To restrict the exploitation of the submission system for other purposes, only submissions with per-task metric values different from zero were accepted as valid, resulting in only 17 complete and valid submissions. In order to avoid overfit but still allow for some degree of methodological development, each team was allowed submit their results 15 times. The winner of the 2018 MSD challenge (nnU-Net, denoted as Isensee on the live challenge), submitted to the live challenge leaderboard on the 6th of December 2019, and held the first position for almost 1 year, until the 30th of October 2020.

Since for the live challenge teams were allowed to tune their method on all ten data sets, the minimum value of the data set specific median DSC improved quite substantially from the 2018 MSD challenge, as shown in Supplementary Fig. 14. The two

hardest tasks during the 2018 MSD challenge were the segmentation of the tumor inside the pancreas, with an overall median of the mean DSC of 0.21 over all participants (0.37 for the top five teams) and the segmentation of the colon cancer primaries, with an overall median of the mean DSC of 0.16 over all participants (0.41 for the top five teams). The worst task for the rolling challenge was the segmentation of the non-enhancing tumor segmentation inside the brain, with a median DSC of 0.47.

At the other end of the spectrum was the spleen segmentation task, where the median task DSC over all participants was 0.94 during the 2018 challenge, and improved to 0.97 for the rolling challenge. These observations suggest that the ability for multiple methods to solve the task has improved, with methods performing slightly better on harder tasks and significantly better on easy tasks.

In 2019 and 2020, the rolling challenges have resulted in three methods that superseded the winning results of the 2018 MSD challenge. Within these two follow-up years, two main trends were observed: the first major trend is the continuous and gradual improvement of "well performing" algorithms, such as the heuristics and task fingerprinting of the nnU-Net method; the second major trend that was observed was the rise of Neural Architecture Search (NAS)[32] among the top teams. More specifically, both the third and the current[33] (as of April 2021) leader of the rolling challenge used this approach. NAS optimizes the network architecture itself to each task in a fully-automated manner. Such an approach uses a network-configuration fitness function that is optimized independently for each task, thus providing an empirical approach for network architectural optimization. When compared to heuristic methods (e.g., nnU-Net), NAS appears to result in improved algorithmic performance at the expense of increased computational cost.

**Conclusion**. Machine learning based semantic segmentation algorithms are becoming increasingly general purpose and accurate, but have historically required significant field-specific expertise to use. The MSD challenge was set up to investigate how accurate fully-automated image segmentation learning methods can be on a plethora of tasks with different types of task complexity. Results from the MSD challenge have demonstrated that fully-automated methods can now achieve state-of-the-art performance without the need for manual parameter optimization, even when applied to previously unseen tasks. A central hypothesis of the MSD challenge—that an algorithm which works well and automatically on several tasks should also work well on other unseen tasks—has been validated among the challenge participants and across tasks. This hypothesis was further corroborated by monitoring the generalizability of the winning method in the 2 years following the challenge, where we found that nnU-Net achieved state-of-the-art performance on many tasks including against task-optimized networks. While it is important to note that many classic semantic segmentation problems (e.g., domain shift and label accuracy) remain, and that methodological progress (e.g., NAS and better heuristics) will continue pushing the boundaries of algorithmic performance and generalizability, the MSD challenge has demonstrated that the training of accurate semantic segmentation networks can now be fully automated. This commoditization of semantic segmentation methods allows computationally-versed scientists that lack AI-specific knowledge to use these techniques without any knowledge on how the models work or how to tune the hyperparameters. However, in order to make the tools easier to use by clinicians and other scientists, the current platforms would need to be wrapped around a graphical user interface and the installation processes need to be made simpler.

## Methods

This section is organized according to the EQUATOR (https://www.equator-network.org) guideline BIAS (Biomedical Image Analysis ChallengeS)[22], a recently published guideline specifically designed for the reporting of biomedical image analysis challenges. It comprises information on challenge organization and mission, as well as the data sets and assessment methods used to evaluate the submitted results.

**Challenge organization**. The Decathlon challenge was organized at the International Conference on Medical Image Computing and Computer Assisted Intervention (MICCAI) 2018, held in Granada, Spain. After the main challenge event at MICCAI, a live challenge was opened for submissions which is still open and regularly receives new submissions (more than 500 as of May 15th 2021).

The MSD challenge aimed to test the ability of machine-learning algorithms to accurately segment a large collection of prescribed regions of interest, as defined by ten different data sets, each corresponding to a different anatomical structure (see Fig. 1) and to at least one medical-imaging task[34]. The challenge itself consisted of two phases:

In the first phase, named the development phase, the training cases (comprising images and labels) for seven data sets were released, namely for brain, liver, heart, hippocampus, prostate, lung, and pancreas. Participants were expected to download the data, develop a general-purpose learning algorithm, train the algorithm on each task's training data independently and without human interaction (no task-specific manual parameter settings), run the learned model on each task's test data, and submit the segmentation results. Each team was only allowed to make one submission per day to avoid model overfit, and the results were presented in form of a live leaderboard on the challenge website (http://medicaldecathlon.com/), visible to the public. Note that participants were only able to see the average performance obtained by their algorithm on the test data of the seven development tasks.

The purpose of the second phase of the challenge, named the mystery phase, was to investigate whether algorithms were able to generalize to unseen segmentation tasks. Teams that submitted to the first phase and completed all necessary steps were invited to download three more data sets (images and labels), i.e., hepatic vessels, colon, and spleen. They were allowed to train their previously developed algorithm on the new data, without any modifications to the method itself. Segmentation results of the mystery phase could only be submitted once. A detailed description of the challenge organization is summarized in is summarized in Appendix A of Supplementary Material, following the form introduced in ref. [22].

**The Decathlon mission**. Medical image segmentation, i.e., the act of labeling or contouring structures of interest in medical-imaging data, is a task of crucial importance, both clinically and scientifically, as it allows the quantitative characterization of regions of interest. When performed by human raters, image segmentation is very time-consuming, thus limiting its clinical usage. Algorithms can be used to automate this segmentation process, but, classically, a different algorithm had to be developed for each segmentation task. The goal of the MSD challenge was finding a single algorithm, or learning system, that would be able to generalize and work accurately across multiple different medical segmentation tasks, without the need for any human interaction.

The tasks of the Decathlon challenge were chosen as a representative sample of real-world applications, so as to test for algorithmic generalizability to these. Different axes of complexity were explicitly explored: the type and number of input modalities, the number of regions of interest, their shape and size, and the complexity of the surrounding tissue environment (see Fig. 1). Detailed information of each data set is provided in "Challenge data sets" and Table 2.

**Challenge data sets**. The Decathlon challenge made ten data sets available online[35], where each data set had between one and three region-of-interest (ROI) targets (17 targets in total). Importantly, all data sets have been released with a permissive copyright-license (CC-BY-SA 4.0), thus allowing for data sharing, redistribution, and commercial usage, and subsequently promoting the data set as a standard test-bed for all users. The images (2,633 in total) were acquired across multiple institutions, anatomies and modalities during real-world clinical applications. All images were de-identified and reformatted to the Neuroimaging Informatics Technology Initiative (NIfTI) format https://nifti.nimh.nih.gov. All images were transposed (without resampling) to the most approximate right-anterior-superior coordinate frame, ensuring the data matrix $x$-$y$-$z$ direction was consistent (using fslreorient2std) and converted to the NIFTI radiological standard. This data transformation was performed to minimize medical-imaging specific data loading issues for non-expert participants. Lastly, non-quantitative modalities (e.g., MRI) were robust min-max scaled to the same range. For each segmentation task, a pixel-level label annotation was provided depending on the definition of each specific task. Information on how the data sets were annotated is provided in[35]. For 8 out of 10 data sets, two thirds of the data were released as training sets (images and labels) and one third as test set (images without labels). As the remaining two tasks (brain tumor and liver) consisted of data from two well-known challenges, the original training/test split was preserved. Note that inter-rater reliability estimates are not available for the MSD tasks due to the complexity

**Table 2 Summary of the ten data sets of the Medical Segmentation Decathlon.**

| Phase | Task | Modality | Protocol | Target | # Cases (Train/Test) |
|---|---|---|---|---|---|
| Development phase | Brain | mp-MRI | FLAIR, T1w, T1 \w Gd, T2w | Edema, enhancing and non-enhancing tumor | 750 4D volumes (484/266) |
| | Heart | MRI | – | Left atrium | 30 3D volumes (20/10) |
| | Hippocampus | MRI | T1w | Anterior and posterior of hippocampus | 394 3D volumes (263/131) |
| | Liver | CT | Portal-venous phase | Liver and liver tumor | 210 3D volumes (131/70) |
| | Lung | CT | – | Lung and lung cancer | 96 3D volumes (64/32) |
| | Pancreas | CT | Portal-venous phase | Pancreas and pancreatic tumor mass | 420 3D volumes (282/139) |
| | Prostate | mp-MRI | T2, ADC | Prostate PZ and TZ | 48 4D volumes (32/16) |
| Mystery phase | Colon | CT | Portal-venous phase | Colon cancer primaries | 190 3D volumes (126/64) |
| | Hepatic Vessels | CT | Portal-venous phase | Hepatic vessels and hepatic tumor | 443 3D volumes (303/140) |
| | Spleen | CT | Portal-venous phase | Spleen | 61 3D volumes (41/20) |

*mp-MRI* multiparametric-magnetic resonance imaging, *FLAIR* fluid-attenuated inversion recovery, *T1w* T1-weighted image, *T1\w Gd* post-Gadolinium (Gd) contrast T1-weighted image, *T2w* T2-weighted image, *CT* computed tomography, *PZ* peripheral zone, *TZ* transition zone.

of double-labeling the data, limiting comparisons to human (or super-human) level performance.

Table 2 presents a summary of the ten data sets, including the modality, image series, ROI targets and data set size. A brief description of each data set is provided below.

- Development Phase (1st) contained seven data sets with thirteen target regions of interest in total:

  1. Brain: The data set consists of 750 multiparametric-magnetic resonance images (mp-MRI) from patients diagnosed with either glioblastoma or lower-grade glioma. The sequences used were native T1-weighted (T1), post-Gadolinium (Gd) contrast T1-weighted (T1-Gd), native T2-weighted (T2), and T2 Fluid-Attenuated Inversion Recovery (FLAIR). The corresponding target ROIs were the three tumor sub-regions, namely edema, enhancing, and non-enhancing tumor. This data set was selected due to the challenge of locating these complex and heterogeneously-located targets. The Brain data set contains the same cases as the 2016 and 2017 Brain Tumor Segmentation (BraTS) challenges[36–38]. The filenames were changed to avoid participants mapping cases between the two challenges.
  2. Heart: The data set consists of 30 mono-modal MRI scans of the entire heart acquired during a single cardiac phase (free breathing with respiratory and electrocardiogram (ECG) gating). The corresponding target ROI was the left atrium. This data set was selected due to the combination of a small training data set with large anatomical variability. The data was acquired as part of the 2013 Left Atrial Segmentation Challenge (LASC)[39].
  3. Hippocampus: The data set consists of 195 MRI images acquired from 90 healthy adults and 105 adults with a non-affective psychotic disorder. T1-weighted MPRAGE was used as the imaging sequence. The corresponding target ROIs were the anterior and posterior of the hippocampus, defined as the hippocampus proper and parts of the subiculum. This data set was selected due to the precision needed to segment such a small object in the presence of a complex surrounding environment. The data was acquired at the Vanderbilt University Medical Center, Nashville, US.
  4. Liver: The data set consists of 201 contrast-enhanced CT images from patients with primary cancers and metastatic liver disease, as a consequence of colorectal, breast, and lung primary cancers. The corresponding target ROIs were the segmentation of the liver and tumors inside the liver. This data set was selected due to the challenging nature of having significant label unbalance between large (liver) and small (tumor) target region of interests (ROIs). The data was acquired in the IRCAD Hôpitaux Universitaires, Strasbourg, France and contained a subset of patients from the 2017 Liver Tumor Segmentation (LiTS) challenge[40].
  5. Lung: The data set consists of preoperative thin-section CT scans from 96 patients with non-small cell lung cancer. The corresponding target ROI was the tumors within the lung. This data set was selected due to the challenge of segmenting small regions (tumor) in an image with a large field-of-view. Data was acquired via the Cancer Imaging.Archive (https://www.cancerimagingarchive.net/).
  6. Prostate: The data set consists of 48 prostate multiparametric MRI (mp-MRI) studies comprising T2-weighted, Diffusion-weighted and T1-weighted contrast-enhanced series. A subset of two series, transverse T2-weighted and the apparent diffusion coefficient (ADC) was selected. The corresponding target ROIs were the prostate peripheral zone (PZ) and the transition zone (TZ). This data set was selected due to the challenge of segmenting two adjoined regions with very large inter-subject variability. The data was acquired at Radboud University Medical Center, Nijmegen Medical Center, Nijmegen, The Netherlands.
  7. Pancreas: The data set consists of 420 portal-venous phase CT scans of patients undergoing resection of pancreatic masses. The corresponding target ROIs were the pancreatic parenchyma and pancreatic mass (cyst or tumor). This data set was selected due to label unbalance between large (background), medium (pancreas) and small (tumor) structures. The data was acquired in the Memorial Sloan Kettering Cancer Center, New York, US.

- Mystery Phase (2nd) contained three (hidden) data sets with four target regions of interest in total:

  1. Colon: The data set consists of 190 portal-venous phase CT scans of patients undergoing resection of primary colon cancer. The corresponding target ROI was colon cancer primaries. This data set was selected due to the challenge of the heterogeneous appearance, and the annotation difficulties. The data was acquired in the Memorial Sloan Kettering Cancer Center, New York, US.
  2. Hepatic Vessels: The data set consists of 443 portal-venous phase CT scans obtained from patients with a variety of primary and metastatic liver tumors. The corresponding target ROIs were the vessels and tumors within the liver. This data set was selected due to the tubular and connected nature of hepatic vessels neighboring heterogeneous tumors. The data was acquired in the Memorial Sloan Kettering Cancer Center, New York, US.
  3. Spleen: The data set consists of 61 portal-venous phase CT scans from patients undergoing chemotherapy treatment for liver metastases. The corresponding target ROI was the spleen. This data set was selected due to the large variations in the field-of-view. The data was acquired in the Memorial Sloan Kettering Cancer Center, New York, US.

**Assessment method**

*Assessment of competing teams.* Two widely known semantic segmentation metrics were used to evaluate the submitted approaches, namely the DSC[9] and the Normalized Surface Distance (NSD)[10], both computed on 3D volumes. The implementation of both metrics can be downloaded in the form of a Jupyter notebook from the challenge website, (http://www.medicaldecathlon.com section Assessment Criteria). A more memory-efficient recently made available implementation of the NSD metric, which has been recently made available, can be obtained by computing the distance transform map using (https://evalutils.readthedocs.io/en/latest/modules.html#evalutils.stats.distance_transform_edt_float32) rather than scipy.ndimage.morphology.distance_transform_edt. The metrics DSC and NSD were chosen due to their popularity, rank stability[34], and smooth, well-understood and well-defined behavior when ROIs do not overlap. Having simple and rank-stable metrics also allows the statistical comparison between methods. For the NSD, tolerance values were based on clinical feedback and consensus, and were chosen by the clinicians segmenting each organ. NSD was defined at task level and was the same for all the targets of each task. The value represented what they would consider an acceptable error for the segmentation they were performing. The following values have been

chosen for the individual tasks (in mm): Brain—5; Heart—4; Hippocampus—1; Liver—7; Lung—2; Prostate—4; Pancreas—5; Colon—4; Hepatic vessel—3; Spleen —3. It is important to note that the proposed metrics are not task-specific nor task-optimal, and thus, they do not fulfill the necessary criteria for clinical algorithmic validation of each task, as discussed in "Challenge assessment".

A so-called significance score was determined for each algorithm $a$, separately for each task/target ROI $c_i$ and metric $m_j \in \{DSC, NSD\}$ and referred to as $s_{i,j}(a)$. Similarly to what was used to infer the ranking across the different BRATS tasks[41], the significance score was computed according to the following four-step process:

1. Performance assessment per case: Determine performance $m_j(a_l, t_{ik})$ of all algorithms $a_l$, with $l = \{1, \ldots, N_A\}$, for all test cases $t_{ik}$, with $k = \{1, \ldots, N_i\}$, where $N_A$ is the number of competing algorithms and $N_i$ is the number of test cases in competition $c_i$. Set $m_j(a_l, t_{ik})$ to 0 if its value is undefined.
2. Statistical tests: Perform a Wilcoxon signed-rank pairwise statistical test between algorithms $(a_l, a_{l'})$, with values $m_j(a_l, t_{ik}) - m_j(a_{l'}, t_{ik})$, $\forall k = \{1, \ldots, N_i\}$.
3. Significance scoring: $s_{i,j}(a_l)$ then equals the number of algorithms performing significantly worse than $a_l$, according to the statistical test (per comparison $\alpha = 0.05$, not adjusted for multiplicity).
4. Significance ranking: The ranking is computed from the scores $s_{i,j}(a_l)$, with the highest score (rank 1) corresponding to the best algorithm. Note that shared scores/ranks are possible. If a task has multiple target ROI, the ranking scheme is applied to each ROI separately, and the final ranking per task is computed as the mean significance rank.

The final score for each algorithm over all tasks of the development phase (the seven development tasks) and over all tasks of the mystery phase (the three mystery tasks) was computed as the average of the respective task's significance ranks. The full validation algorithm was defined and released prior to the start of the challenge, and available on the decathlon website (http://medicaldecathlon.com/files/MSD-Ranking-scheme.pdf).

To investigate ranking uncertainty and stability, bootstrapping methods were applied with 1000 bootstrap samples as described in[34]. The statistical analysis was performed using the open-source R toolkit challengeR (https://phabricator.mitk.org/source/challenger/), version 1.0.2[17], for analyzing and visualizing challenge results. The original rankings computed for the development and mystery phases were compared to the ranking lists based on the individual bootstrap samples. The correlation of pairwise rankings was determined via Kendall's $\tau$[18], which provides values between −1 (for reverse ranking order) and 1 (for identical ranking order). The source code for generating the results presented in "Results" and the Appendix is publicly available (https://phabricator.mitk.org/source/msd_evaluation/).

*Monitoring of the challenge winner and algorithmic progress.* To investigate our hypothesis that a method capable of performing well on multiple tasks will generalize its performance to an unseen task, and potentially even outperform a custom-designed task-specific solution, we monitored the winner of the challenge for a period of 2 years. Specifically, we reviewed the rank analysis and leaderboards presented in the corresponding article[8], as well as the leaderboard of challenges from the grand-challenge.org website organized in 2020. We also reviewed further articles mentioning the new state-of-the-art method nnU-Net[19]. Finally, as the MSD challenge submission was reopened after the challenge event (denoted the "MSD Live Challenge"), we monitored submissions for new algorithmic approaches which achieve state-of-the-art performance, in order to probe new areas of scientific interest and development.

**Reporting summary**. Further information on research design is available in the Nature Research Reporting Summary linked to this article.

## Data availability

*Challenge data set*. The MSD data set is publicly available under a Creative Commons license CC-BY-SA4.0, allowing broad (including commercial) use. The training data used in this study is available at http://medicaldecathlon.com/. The test data of the challenge cannot be released since the live challenge is still open and users are able to submit their results anytime; we currently have no intentions of closing the challenge.

*Challenge assessment data*. The raw challenge assessment data used to calculate the challenge rankings cannot be made publicly available due to privacy reasons. It contains the DSC and NSD values for every participating team for every task and target region. However, the aggregated results can be found in Table 1 and Supplementary Tables 2–11. Furthermore, they can be found here: https://phabricator.mitk.org/source/msd_evaluation/ in the folders descriptive-statistics, mean-values-per-subtask and rankings-per-subtask.

## Code availability

The implementation of the metrics used in the challenge, namely the DSC and NSD, were provided as a Python Notebook[42]. The significance rankings have been computed

with the R package challengeR, version 1.0.2, which is publicly available: https://phabricator.mitk.org/source/challenger/. Finally, the code to compute the final rankings and all tables and figures of this paper can be found here: https://phabricator.mitk.org/source/msd_evaluation/.

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

## Acknowledgements

This work was supported by the UK Research and Innovation London Medical Imaging & Artificial Intelligence Center for Value-Based Healthcare. Investigators received support from the Wellcome/EPSRC Center for Medical Engineering (WT203148), Wellcome Flagship Program (WT213038). The research was also supported by the Bavarian State Ministry of Science and the Arts and coordinated by the Bavarian Research Institute for Digital Transformation and by the Helmholtz Imaging Platform (HIP), a platform of the Helmholtz Incubator on Information and Data Science. Team CerebriuDIKU gratefully acknowledges support from the Independent Research Fund Denmark through the project U-Sleep (project number 9131-00099B). R.M.S. is supported by the Intramural Research Program of the National Institutes of Health Clinical Center G.L. reported research grants from the Dutch Cancer Society, the Netherlands Organization for Scientific Research (NWO), and HealthHolland during the conduct of the study, and grants from Philips Digital Pathology Solutions, and consultancy fees from Novartis and Vital Imaging, outside the submitted work. Research reported in this publication was partly supported by the National Institutes of Health (NIH) under award numbers NCI:U01CA242871, NCI:U24CA189523, NINDS:R01NS042645. The content of this publication is solely the responsibility of the authors and does not represent the official views of the NIH.Henkjan Huisman is receiving grant support from Siemens Healthineers. James Meakin received grant funding from AWS. The method presented by BCVUniandes was made in collaboration with Silvana Castillo, from Universidad de los Andes. We would like to thank Minu D. Tizabi for proof-reading the paper.

## Author contributions
M.A. worked on the conceptual design and data preparation of the challenge, gave challenge day-to-day support as co-organizer, coordinated the work, validated the participating methods and wrote the document. A.R. worked on the conceptual design of the challenge, coordinated the work, performed the statistical analysis of participating methods, designed the figures and wrote the document. S.B. donated the brain tumor data set and co-organized the challenge. K.F. donated the lung tumors data set and co-organized the challenge. A.K.S. worked on the conceptual design of the challenge, led the statistical analysis committee and co-organized the challenge. B.A.L. worked on the conceptual design, was a member of the metrics committee, co-organized the challenge and donated the hippocampus data set. G.L. donated the prostate data set and co-organized the challenge. B.M. donated the brain and liver tumors data sets for the challenge, was a member of the statistics and metrics committee and co-organized the challenge. O.R. worked on the conceptual design of the challenge, was a member of the metrics committee and co-organized the challenge. R.M.S. worked on the conceptual design and co-organized the challenge. B.v.G. worked on the conceptual design, co-organized the challenge and donated the prostate data set. A.L.S. worked on the conceptual design, co-organized the challenge and donated the pancreas, colon cancer, hepatic vessels and spleen data sets for the challenge. M.B., P.B., P.F.C., R.K.G.D., M.J.G., S.H.H., H.H., W.R.J., M.K.M., S.N., J.S.G.P., K.R., C.T.G., and E.V. donated data for the challenge. H.H. and J.A.M. supported the grand-challenge.org submissions. SO co-organized the challenge. M.W. implemented the toolkit for the statistical ranking analysis for the challenge. P.A., B.B., S.C., L.D., J.F., B.H., F.I., Y.J., F.J., N.K., I.K., D.M., A.P., B.P., M.P., R.R., O.R., I.S., W.S., J.S., C.W., L.W., Y.W., Y.X., D.X., Z.X., and Y.Z. participated in the challenge. LMH initiated and co-organized the challenge, worked on the conceptual design and was a member of the statistical analysis committee, coordinated the work and wrote the document. M.J.C. initiated and organized the challenge (lead), coordinated the work and wrote the document.

## Competing interests
No funding contributed explicitly to the organization and running of the challenge. The challenge award has been kindly provided by NVIDIA. However, NVIDIA did not influence the design or running of the challenge as they were not part of the organizing committee. R.M.S. received royalties from iCAD, Philips, ScanMed, Translation Holdings, and PingAn. Individually funding sourcing unrelated to the challenge has been listed in the Acknowledgments section. The remaining authors declare no competing interests.

## Additional information

[1]School of Biomedical Engineering & Imaging Sciences, King's College London, London, UK. [2]Div. Computer Assisted Medical Interventions, German Cancer Research Center (DKFZ), Heidelberg, Germany. [3]HI Helmholtz Imaging, German Cancer Research Center (DKFZ), Heidelberg, Germany. [4]Faculty of Mathematics and Computer Science, University of Heidelberg, Heidelberg, Germany. [5]Center for Biomedical Image Computing and Analytics (CBICA), University of Pennsylvania, Philadelphia, PA, USA. [6]Department of Radiology, Perelman School of Medicine, University of Pennsylvania, Philadelphia, PA, USA. [7]Department of Pathology and Laboratory Medicine, Perelman School of Medicine, University of Pennsylvania, Philadelphia, PA, USA. [8]Center for Biomedical Informatics and Information Technology, National Cancer Institute (NIH), Bethesda, MD, USA. [9]Div. Biostatistics, German Cancer Research Center (DKFZ), Heidelberg, Germany. [10]Electrical Engineering and Computer Science, Vanderbilt University, Nashville, TN, USA. [11]Radboud University Medical Center, Radboud Institute for Health Sciences, Nijmegen, The Netherlands. [12]Quantitative Biomedicine, University of Zurich, Zurich, Switzerland. [13]DeepMind, London, UK. [14]Imaging Biomarkers and Computer-Aided Diagnosis Laboratory, Department of Radiology and Imaging Sciences, National Institutes of Health Clinical Center (NIH), Bethesda, MD, USA. [15]Department of Informatics, Technische Universität München, München, Germany. [16]Department of Radiology, Memorial Sloan Kettering Cancer Center, New York, NY, USA. [17]Department of Psychiatry & Behavioral Sciences, Vanderbilt University Medical Center, Nashville, TN, USA. [18]Department of Surgery, Memorial Sloan Kettering Cancer Center, New York, NY, USA. [19]Department of Radiology, Stanford University, Stanford, CA, USA. [20]Department of Computer Science and Software Engineering, École Polytechnique de Montréal, Montréal, QC, Canada. [21]Universidad de los Andes, Bogota, Colombia. [22]VUNO Inc., Seoul, Korea. [23]Tencent Jarvis Lab, Shenzhen, China. [24]Department of Automation, Tsinghua University, Beijing, China. [25]Shenzhen Institute of Advanced Technology, Chinese Academy of Sciences, Shenzhen, China. [26]HI Applied Computer Vision Lab, Division of Medical Image Computing, German Cancer Research Center (DKFZ), Heidelberg, Germany. [27]Department of Computer Science, Xiamen University, Xiamen, China. [28]Kakao Brain, Seongnam-si, Republic of Korea. [29]Cerebriu A/S, Copenhagen, Denmark. [30]Pattern Analysis and Learning Group, Department of Radiation Oncology, Heidelberg University Hospital, Heidelberg, Germany. [31]Institute of Imaging & Computer Vision, RWTH Aachen University, Aachen, Germany. [32]Fraunhofer Institute for Digital Medicine MEVIS, Bremen, Germany. [33]Department of Computer Science, University of Copenhagen, Copenhagen, Denmark. [34]MaaDoTaa.com, San Diego, CA, USA. [35]Lab for Artificial Intelligence in Medical Imaging (AI-Med), Department of Child and Adolescent Psychiatry, University Hospital, LMU München, Germany. [36]MoE Key Lab of Artificial Intelligence, AI Institute, Shanghai Jiao Tong University, Shanghai, China. [37]Shanghai Key Laboratory of Multidimensional Information Processing, East China Normal University, Shanghai, China. [38]Johns Hopkins University, Baltimore, MD, USA. [39]NVIDIA, Santa Clara, CA, USA. [40]School of Computing/Department of Biomedical and Molecular Sciences, Queen's University, Kingston, ON, Canada. [41]Medical Faculty, University of Heidelberg, Heidelberg, Germany. [42]These authors contributed equally: Michela Antonelli, Annika Reinke. [43]These authors jointly supervised this work: Lena Maier-Hein, M. Jorge Cardoso. ✉email: michela.antonelli@kcl.ac.uk

