## [Peer Review File · Nature Communications]

Reviewers' Comments:

Reviewer #1:

Remarks to the Author:

This paper presents a large-scale and diverse medical image segmentation dataset and an international challenge was well designed to evaluate the algorithm generalizability.

The well-known nnU-Net was born in this challenge, which has demonstrated that the algorithm can generalize on (unseen) different segmentation tasks and preserve good average performance. The paper is well-written and easy to follow. There is little to no technical novelty but a well-designed challenge can be the fertile soil for methodology development.

Some sub-datasets have critical issues that should be fixed. The specific comments are provided as follows.

Abstract:

I think the third conclusion “the training of accurate AI segmentation models is now commoditized to non AI experts.” is exaggerated, because using nnUNet requires basic deep learning and python expertise. If checking the issues in nnUNet Github repository, you can find many usage questions. Moreover, directly pip install nnunet can be very slow during training. In order to get maximum performance, one should manually compile pytorch with a recent cuDNN version (8002 or newer is a must!). Thus, I do not think non AI experts (e.g., without python expertise) can easily train the AI model. If the author can provide a graphical pipeline to train the SOTA AI model, I would agree with this claim.

Sec. 2.3 Challenge data sets

Page 7. Could you please make your pre-processing code (e.g., reformatted, transposed to RAS) publicly available?

Brain dataset: Please specify the difference to BraTS16-17 dataset. E.g, how many cases belong to BraTS. Besides, the cases are from 19 institutions. Could you please provide the medical center information for each case by a e.g., csv/xlsx file?

Liver dataset: following cases missed the correct spacing information.

liver_28.nii.gz spacing: (1.0, 1.0, 1.0)

liver_29.nii.gz spacing: (1.0, 1.0, 1.0)

liver_30.nii.gz spacing: (1.0, 1.0, 1.0)

liver_31.nii.gz spacing: (1.0, 1.0, 1.0)

liver_32.nii.gz spacing: (1.0, 1.0, 1.0)

liver_33.nii.gz spacing: (1.0, 1.0, 1.0)

liver_34.nii.gz spacing: (1.0, 1.0, 1.0)

liver_35.nii.gz spacing: (1.0, 1.0, 1.0)

liver_36.nii.gz spacing: (1.0, 1.0, 1.0)

liver_37.nii.gz spacing: (1.0, 1.0, 1.0)

liver_38.nii.gz spacing: (1.0, 1.0, 1.0)

liver_39.nii.gz spacing: (1.0, 1.0, 1.0)
liver_40.nii.gz spacing: (1.0, 1.0, 1.0)
liver_41.nii.gz spacing: (1.0, 1.0, 1.0)
liver_42.nii.gz spacing: (1.0, 1.0, 1.0)
liver_43.nii.gz spacing: (1.0, 1.0, 1.0)
liver_44.nii.gz spacing: (1.0, 1.0, 1.0)
liver_45.nii.gz spacing: (1.0, 1.0, 1.0)
liver_46.nii.gz spacing: (1.0, 1.0, 1.0)
liver_47.nii.gz spacing: (1.0, 1.0, 1.0)

Moreover, Could you please provide the medical center information for each case?

Pancreas dataset: I re-download the dataset and find that the number of scans is still 420 (281+139) rather than 421.

Hepatic Vessel dataset: The vessel labels are too noisy. Many vessels are not annotated.

Could you please refine the vessel annotation?

Here is a good example <https://www.ircad.fr/research/3dircadb/>

Common issues:

- 1) please provide the vendor/ manufacturer information for each dataset.
- 2) please give a brief introduction about how the scans were annotated.
- 3) please delete the .DS_Store file in the dataset.

2.4 Assessment method

- 1) The tolerance of NSD in each task must be presented because it's important for users to understand the segmentation results.
- 2) Regarding the http://medicaldecathlon.com/files/Surface_distance_based_measures.ipynb, it would be more memory-efficient if computing the distance transform map by https://evalutils.readthedocs.io/en/latest/modules.html#evalutils.stats.distance_transform_edt_float32 rather than `scipy.ndimage.morphology.distance_transform_edt`.
- 3) It's great to make the metric code and statistical analysis code public available. Could you please make the "significance score" computation code publicly available as well?

3.1 Challenge submissions

180 teams registered for the challenge but only 19 teams submitted final and valid results for the mystery phase. Why 89% teams did not make final submissions? Are there any suggestions to improve the submission rate?

5.2 Challenge data set

It would be better to provide inter-rater segmentation performance for each task. For example, the author can compute this score by inviting 3-5 radiologists to annotate 5-10 testing cases.

5.3 Challenge assessment

How do you choose the proper tolerance in NSD?

Reviewer #2:

Remarks to the Author:

The paper is very interesting for it not only describes a very challenging dataset but it also reviews models that could cope with it. The presented statistics are interesting and are significant in the field, for they confirm that the segmentation of medical images can be effectively handled by different versions of UNet.

The analysis is sound and convincing and supported by the methods' results.

The paper is well organized and well written.

There is only one information I couldn't find, which regards the code availability. Do the authors of the winning methods provide the code somewhere? I think this is an information that should be inserted in Table 2, where the methods details are reported

Dear Editor,

Many thanks for reviewing our paper “The Medical Segmentation Decathlon” (ref: NCOMMS-21-32431-T). We have revised the manuscript in line with the reviewers’ comments. For your convenience, in this response letter, we have included reviewers’ remarks (in italics) followed by our responses (in plain text). Line numbers are referred to the clean version of the manuscript.

Best regards,

Michela Antonelli

Reviewer #1

This paper presents a large-scale and diverse medical image segmentation dataset and an international challenge was well designed to evaluate the algorithm generalizability. The well-known nnU-Net was born in this challenge, which has demonstrated that the algorithm can generalize on (unseen) different segmentation tasks and preserve good average performance. The paper is well-written and easy to follow. There is little to no technical novelty but a well-designed challenge can be the fertile soil for methodology development. Some sub-datasets have critical issues that should be fixed. The specific comments are provided as follows.

Abstract:

- 1) *I think the third conclusion “the training of accurate AI segmentation models is now commoditized to non AI experts.” is exaggerated, because using nnUNet requires basic deep learning and python expertise. If checking the issues in nnUNet Github repository, you can find many usage questions. Moreover, directly pip install nnunet can be very slow during training. In order to get maximum performance, one should manually compile pytorch with a recent cuDNN version (8002 or newer is a must!). Thus, I do not think non AI experts (e.g., without python expertise) can easily train the AI model. If the author can provide a graphical pipeline to train the SOTA AI model, I would agree with this claim.*

We thank the reviewer for this remark and softened the sentence to “scientists that are not versed in AI model training” instead of non AI experts.

We believe it is important to preserve the point that scientists can now use certain algorithms/frameworks without deep knowledge of how the AI algorithms work or how to best choose parameters. As stated by the reviewer, indeed, users of such a tool would still need to be technically versed, as one would still need to format the data appropriately and run the scripts using the command line. We agree that creating a graphical user interface for some of the proposed methods would be a great asset, and we will contact the developers of the methods to propose they develop such a tool. As the abstract is word limited, we added the following paragraph to the conclusion of the paper on page 21.

"This commoditization of semantic segmentation methods allows computationally-versed scientists that lack AI-specific knowledge to use these techniques without any knowledge on how the models work or how to tune the hyperparameters. However, in order to make the tools easier to use by clinicians and other scientists, the current platforms would need to be wrapped around a graphical user interface and the installation processes need to be made simpler."

Sec. 2.3 Challenge data sets

- 1) *Page 7. Could you please make your pre-processing code (e.g., reformatted, transposed to RAS) publicly available?*

To pre-process data we used FSL library functions to re-orient, rescale and normalize images. As per the text below, even though most images were transformed using `fslreorient2std` to correct for ill-defined headers, and `fsl_maths` to correct for other issues (e.g., Nan present in images). In order to simplify data usage and avoid mismatches between the XYZ millimeter coordinates and the IJK data matrix coordinates between subjects, images had to undergo image-specific corrections (conversion from neurological to radiological standard, nans outside of the field of view were removed and replaced with the image background intensity, header intensity slope/intercept were applied to ensure a match between the stored voxel value and the quantitative value, etc) when problems were identified during the quality control phase. The following paragraph has been added on page 32.

"To pre-process data we used FSL library functions. In particular, all images were transposed (without resampling) to the most approximate right-anterior-superior coordinate frame, ensuring the data matrix x-y-z direction was consistent using `fslreorient2std`. Lastly, non-quantitative modalities (e.g., MRI) were robust min-max scaled to the same range by means of a mixture of `fsl_maths` and `fsl_stats`."

Note that while most images were processed automatically, some others images, due to their ill-defined DICOM headers had to undergo specific processing to correct for orientation problems. This pre-processing was done to minimise the burden on participants regarding medical-imaging specific data loading issues. We further clarified the transformations in the paper and added the following text to clarify the need for this transformation:

"This data transformation was performed to minimise medical-imaging specific data loading issues for non-expert participants"

- 2) *Brain dataset: Please specify the difference to BraTS16-17 dataset. E.g, how many cases belong to BraTS. Besides, the cases are from 19 institutions. Could you please provide the medical center information for each case by a e.g., csv/xlsx file?*

Brain dataset contains the same cases as BraTS16-17, but we changed the filenames to avoid participants mapping cases between the two challenges as requested by BraTS challenge authors. For this reason, we do not have the medical centre information for each case. However, we would like to note that even if this information was available, the Decathlon challenge was organised with the commitment that no other information was going to be available about the datasets to minimise discrepancies between tasks. Hence, the datasets were randomized and the random key discarded so that participants (or even organisers) could not use site information as an extra input variable.

We have clarified this point by adding the following paragraph to sub-section "Challenge data sets" on page 33:

"The Brain dataset contains the same cases as the 2016 and 2017 Brain Tumor Segmentation (BraTS) challenges[12,13,14]. The filenames were changed to avoid participants mapping cases between the two challenges."

Liver dataset: following cases missed the correct spacing information.

liver_28.nii.gz spacing: (1.0, 1.0, 1.0) liver_29.nii.gz spacing: (1.0, 1.0, 1.0)

liver_30.nii.gz spacing: (1.0, 1.0, 1.0) liver_31.nii.gz spacing: (1.0, 1.0, 1.0)

liver_32.nii.gz spacing: (1.0, 1.0, 1.0) liver_33.nii.gz spacing: (1.0, 1.0, 1.0)

liver_34.nii.gz spacing: (1.0, 1.0, 1.0) liver_35.nii.gz spacing: (1.0, 1.0, 1.0)

liver_36.nii.gz spacing: (1.0, 1.0, 1.0) liver_37.nii.gz spacing: (1.0, 1.0, 1.0)

liver_38.nii.gz spacing: (1.0, 1.0, 1.0) liver_39.nii.gz spacing: (1.0, 1.0, 1.0)

liver_40.nii.gz spacing: (1.0, 1.0, 1.0) liver_41.nii.gz spacing: (1.0, 1.0, 1.0)

liver_42.nii.gz spacing: (1.0, 1.0, 1.0) liver_43.nii.gz spacing: (1.0, 1.0, 1.0)

liver_44.nii.gz spacing: (1.0, 1.0, 1.0) liver_45.nii.gz spacing: (1.0, 1.0, 1.0)

liver_46.nii.gz spacing: (1.0, 1.0, 1.0) liver_47.nii.gz spacing: (1.0, 1.0, 1.0)

We would like to note that, apart from a header reorientation to an RAS coordinate frame, no other transformations were done to the LiTS data. All the images in the liver dataset were reused from the 2017 Liver Tumor Segmentation (LiTS) challenge, thus any header issues in the original dataset would permeate to the decathlon task. We do not have access to the original LiTS DICOM data, so we, unfortunately, cannot confirm if the spacing is incorrect or not. Note, however, that algorithms performed well on these images; if the image spacing is indeed incorrect, this would mean that algorithms are resilient to possible mis-conversion.

3) *Moreover, Could you please provide the medical center information for each case?*

At the time of challenge design, the organizers decided that only the imaging data would be provided to avoid other variables (such as the use of external information like site or patient details) being a contributing factor to algorithmic performance. If this information

was made available, such information could be exploited by teams by giving this extra input information to models, or by applying techniques such as domain adaptation, making the comparison between algorithms more complicated and dependent on confounds. Lastly, one should also note that because the challenge comprises 10 tasks and because this information was only available for a subset of the tasks, releasing such information only for a subset of the tasks would make task performance comparison complex (as we would not be able to distinguish if a certain algorithm is better on a task or just better at exploiting the extra information). Because of this, a decision was made prior to the challenge data release date, to not make this information available.

- 4) *Pancreas dataset: I re-download the dataset and find that the number of scans is still 420 (281+139) rather than 421.*

We thank the reviewer for spotting this important issue. One case was removed from the Pancreas dataset during the quality control phase due to missing slices (likely caused by an incomplete DICOM set). We updated the number on page 34 and in Figure 1.

- 5) *Hepatic Vessel dataset: The vessel labels are too noisy. Many vessels are not annotated. Could you please refine the vessel annotation? Here is a good example <https://www.ircad.fr/research/3dircadb/>*

We agree with the reviewer that the quality of the vessel annotations is not optimal. We followed the common procedure of having one expert annotator per case for this data set. In line with best practice recommendations on challenges [Maier-Hein et al. 2018, Maier-Hein et al. 2020, Reinke et al. 2018] we further adhere to not changing the challenge design after it has been released to the participants. Importantly, however, the message of the manuscript would not change at all when omitting the vessel data set from the competition. It is also important to note that labels can often be non-optimal from an accuracy point of view, so resilience to non-optimal labels is likely an important trait for AutoML models. To address the reviewers' comment, we have added the following paragraph to the limitation section on page 16:

“Finally, one specific region from one task (the vessel annotations of liver dataset) was found to be non-optimal from a segmentation point of view after the data release, we opted to follow the best practice recommendations on challenges \citep{MaierHein2018, maier2020bias, reinke2018exploit} and not change the challenge design after it was released to participants. Note, however, that the message of this challenge would not change if the vessel data set was omitted from the competition. “

- 6) *Common issues:*

- a) *please provide the vendor/ manufacturer information for each dataset.*

Unfortunately, we can not release this information. We made this challenge design choice to avoid that participants would use this information to

train/optimize their models, as such information could be used in a domain adaptation setup, and would create an information unbalance between tasks.

b) *please give a brief introduction about how the scans were annotated.*

Further details on the annotations have been provided in the corresponding MSD data paper [Simpson et al. 2019]. Note that the current paper is not focusing on the data itself, but rather on the challenge results.

We have added the following paragraph in sub-section "challenge dataset" on page 33:

"For each segmentation task, a pixel-level label annotation was provided depending on the definition of each specific task. Information on how the cases were annotated is provided in [11]."

c) *please delete the .DS_Store file in the dataset.*

We thank the reviewer for this comment; we have removed all the files from each dataset.

2.4 Assessment method

1) *The tolerance of NSD in each task must be presented because it's important for users to understand the segmentation results.*

We thank the reviewer for making this important point. The NSD tolerances are based on clinical feedback and consensus and were chosen by the clinicians responsible for segmenting each organ. The value represented what they would consider an acceptable error for the segmentation they were performing. The following thresholds were chosen (in mm):

Brain: 5mm; Heart: 4; Hippocampus: 1mm; Liver: 7mm; Lung: 2mm; Prostate: 4mm; Pancreas: 5; Colon: 4mm; Hepaticvessel: 3mm; Spleen: 3mm

We added the following paragraph in section "Assessment of competing teams" on page 37:

"For the NSD, tolerance values were based on clinical feedback and consensus and were chosen by the clinicians segmenting each organ. The value represented what they would consider an acceptable error for the segmentation they were performing. The following values have been chosen for the individual tasks (in mm): Brain - 5; Heart - 4; Hippocampus - 1; Liver - 7; Lung - 2; Prostate - 4; Pancreas - 5; Colon - 4; Hepaticvessel - 3; Spleen - 3."

2) *Regarding the*

http://medicaldecathlon.com/files/Surface_distance_based_measures.ipynb, it would be more memory-efficient if computing the distance transform map by https://evalutils.readthedocs.io/en/latest/modules.html#evalutils.stats.distance_transform_edt_float32 rather than `scipy.ndimage.morphology.distance_transform_edt`.

Based on the comment of the reviewer, we added the following footnote to section 2.4.1 ' on page 36: "A more memory-efficient recently implementation of the NSD metric, which has been recently made available, can be obtained by computing the distance transform map using `\url{https://evalutils.readthedocs.io/en/latest/modules.html#evalutils.stats.distance_transform_edt_float32}` rather than `\url{scipy.ndimage.morphology.distance_transform_edt}`"

- 3) *It's great to make the metric code and statistical analysis code public available. Could you please make the "significance score" computation code publicly available as well?*

The significance scores have been generated for every target region with the R package challengeR (<https://phabricator.mitk.org/source/challenger/>), using the 'testBased' ranking scheme. We now further published the code to aggregate the individual rankings to a final ranking: https://phabricator.mitk.org/source/msd_evaluation/ .

This repository further contains the code to calculate other results, tables and figures presented in the paper. Unfortunately, due to privacy (and potential cheating) reasons we are not allowed to publish the raw data containing the challenge participants metric scores on single images. But the repository contains the aggregated results (mean metric scores per algorithm and region, descriptive statistics, rankings per region and final rankings).

This repository is now mentioned in the new section 'Code availability' on page 23.

3.1 Challenge submissions

- 1) *180 teams registered for the challenge but only 19 teams submitted final and valid results for the mystery phase. Why 89% teams did not make final submissions? Are there any suggestions to improve the submission rate?*

We agree with the reviewer. In fact, this is a very typical phenomenon for challenges in the biomedical community. For example, in a survey sent to all MICCAI 2021 challenge organizers, the median of registered participants was 125, but only a median of 17 teams actually participated (see <https://youtu.be/pACDviHdHMI> for reference).

This problem may have multiple reasons: Some challenge participants are only interested in downloading the data (MSD participants likely did not realise that registration was unnecessary for download). But more often, challenge participants are not able to submit their methods within the deadline due to other commitments. Finally, it may happen that participants are not satisfied with their training and validation performance and step back from the final submission.

We added a paragraph in the Discussion section on page 17:

“A total of 180 teams registered for the MSD challenge, of which only 31 teams submitted valid results for the development and 19 teams for the mystery phase. Having a high number of registrations but only a fraction of final participants is a typical phenomenon happening for biomedical image analysis challenges (e.g. the Skin lesion analysis toward melanoma detection 2017 challenge with 46/593 submissions, the Robust Medical Instrument Segmentation (RobustMIS) challenge 2019 with 12/75 submissions or the Multi-Centre, Multi-Vendor and Multi-Disease Cardiac Segmentation (M&Ms) challenge 2020 with 16/80 submissions). Many challenge participants usually register to get data access. However, teams are often not able to submit their methods within the deadline due to other commitments. Furthermore, participants may be dissatisfied with their training and validation performance and step back from the final submission.”

5.2 Challenge data set

- 1) *It would be better to provide inter-rater segmentation performance for each task. For example, the author can compute this score by inviting 3-5 radiologists to annotate 5-10 testing cases.*

We agree with the reviewer; it would have been ideal to provide inter-rater variability for all tasks. Unfortunately, as much of the data has been reused, historical, and labeled by different experts using different labeling mechanism, a relabeling of even a small subset of the data using the original protocols was deemed too complex. We have had initial discussions with RSNA to attempt to relabel the full Decathlon dataset using the RSNA labeling platform and RSNA-associated radiologists, but this is likely to be a multi-year effort.

Nonetheless, we have added a disclaimer in the paper to address this important point by the following new text in the manuscript on page 33 :

“Note that inter-rater reliability estimates are not available for the MSD tasks due to the complexity of double-labeling the data, limiting comparisons to human (or super-human) level performance.”

5.3 Challenge assessment

- 1) *How do you choose the proper tolerance in NSD ?*

As we stated in response to question 2.4.1, the NSD tolerances are based on clinical opinions and were chosen by the clinicians segmenting each organ. The value represented what they would consider an acceptable error for the segmentation they were performing.

Reviewer #2

The paper is very interesting for it not only describes a very challenging dataset but it also reviews models that could cope with it. The presented statistics are interesting and are significant in the field, for they confirm that the segmentation of medical images can be effectively handled by different versions of UNet.

*The analysis is sound and convincing and supported by the methods' results.
The paper is well organized and well written.*

There is only one information I couldn't find, which regards the code availability. Do the authors of the winning methods provide the code somewhere? I think this is an information that should be inserted in Table 2, where the methods details are reported

We thank the reviewer for making this important point! We have added in the footnotes of Table 2 the link to the code for the methods that made it available.

References

[Maier-Hein et al. 2018] Maier-Hein, Lena, et al. "Why rankings of biomedical image analysis competitions should be interpreted with care." *Nature communications* 9.1 (2018): 1-13.

[Reinke et al. 2018] Reinke et al. "How to exploit weaknesses in biomedical challenge design and organization." *International Conference on Medical Image Computing and Computer-Assisted Intervention*. Springer, Cham, 2018.

[Maier-Hein et al. 2020] Maier-Hein, Lena, et al. "BIAS: Transparent reporting of biomedical image analysis challenges." *Medical image analysis* 66 (2020): 101796.

[Simpson et al. 2019] Simpson, Amber L., et al. "A large annotated medical image dataset for the development and evaluation of segmentation algorithms." *arXiv preprint arXiv:1902.09063* (2019).

Reviewers' Comments:

Reviewer #1:

Remarks to the Author:

Thanks for the revision. The authors have addressed most of the comments but I think two important points should be further enhanced.

1. For the low-quality labels in the hepatic vessel dataset, the authors adhere to not changing the challenge design after it has been released to the participants. It's OK to keep the current datasets. However, the authors can also release the refined datasets, which can further enhance the impact of the dataset. For example, participants can develop not only generalized models (for the challenge) but also tailored models for each specific dataset.

Thus, I highly recommend the authors revise the ground truth of the hepatic vessel dataset.

2. Regarding the inter-rater segmentation variability, the authors claimed that it was deemed too complex because the datasets were labeled by different experts using different labeling mechanisms. I agree that it is hard to find the original annotators to re-label the cases following the original annotation protocols. As an alternative, the authors can invite 3-5 new raters to annotate few cases based on typical annotation protocols. Although the protocols could be different from the original protocols, the annotations are still very valuable because it reflects the real inter-rater variability. Since several authors are from medical-related organizations, it's not difficult to find the raters.

I understand that it needs more effort to fix both two points. However, all the two points are doable. This paper is also publishable if the two points are not addressed, but the journal "Medical Image Analysis" would be a more appropriate avenue.

Minor:

Page 37: Do you use the same tolerance values for different targets in each task? For example, the brain tumor segmentation task has three targets. Does the same tolerance value (5mm) apply to all three targets?

typo: blueFor the NSD,tolerance

Dear Editor,

Many thanks for reviewing our paper “The Medical Segmentation Decathlon” (ref: NCOMMS-21-32431-T). We have revised the manuscript in line with the reviewers’ comments. For your convenience, in this response letter, we have included reviewers’ remarks (in italics) followed by our responses (in plain text). Line numbers are referred to the clean version of the manuscript.

Best regards,

Michela Antonelli

Reviewer #1

Thanks for the revision. The authors have addressed most of the comments but I think two important points should be further enhanced.

1. For the low-quality labels in the hepatic vessel dataset, the authors adhere to not changing the challenge design after it has been released to the participants. It's OK to keep the current datasets. However, the authors can also release the refined datasets, which can further enhance the impact of the dataset. For example, participants can develop not only generalized models (for the challenge) but also tailored models for each specific dataset.

Thus, I highly recommend the authors revise the ground truth of the hepatic vessel dataset.

Although we agree with the reviewer and appreciate that releasing a new refined hepatic vessel dataset would be beneficial for the medical image community, preparing such data would require too much time and resource that, at the moment, we do not have. Moreover, it would delay the paper publication without adding any added value to the paper itself.

2. Regarding the inter-rater segmentation variability, the authors claimed that it was deemed too complex because the datasets were labeled by different experts using different labeling mechanisms. I agree that it is hard to find the original annotators to re-label the cases following the original annotation protocols. As an alternative, the authors can invite 3-5 new raters to annotate few cases based on typical annotation protocols. Although the protocols could be different from the original protocols, the annotations are still very valuable because it reflects the real inter-rater variability. Since several authors are from medical-related organizations, it's not difficult to find the raters.

Please note that the MSD challenge is using historical data, and thus, we do not have the resources at the moment to re-label the data, even partially. Also note that the labelling tools used for each dataset are dataset-specific (some using PACS and some using research grade tools such as ITK-Snap), meaning that reproducing the same labelling framework and environment is very challenging. For this reason, we have started a collaboration with RSNA to make use of their new platform to re-label all the data in a consistent way. However, this endeavour is likely going to take more than 1 year, meaning that it would not be ready for the MSD paper. Also note that inter-rater variability was not proposed during the design phase of the challenge, which according to the Bias guidelines means it should not be introduced post-factum to avoid bias.

Minor:

Page 37: Do you use the same tolerance values for different targets in each task? For example, the brain tumor segmentation task has three targets. Does the same tolerance value (5mm) apply to all three targets?

Correct, the tolerance is at task level, it is the same for all the targets of each task. We have clarified this point on page 37 adding the following paragraph:

“NSD was defined at task level and was the same for all the targets of each task.”

typo: blueFor the NSD,tolerance

We thank the reviewer to spot this typo. It has been amended in the new version of the paper.